# Experimental Results and Performance Analysis of a 1 × 2 × 1 UHF MIMO Passive RFID System

**DOI:** 10.3390/s21186308

**Published:** 2021-09-21

**Authors:** Helio Augusto Muzamane, Hsin-Chin Liu

**Affiliations:** Departments of Electrical Engineering, National Taiwan University of Science and Technology, Taipei City 106, Taiwan; d10607804@mail.ntust.edu.tw

**Keywords:** backscatter, passive RFID, UHF, MIMO

## Abstract

Ultra-high frequency (UHF) multiple input multiple output (MIMO) passive radio frequency identification (RFID) systems have attracted the attention of many researchers in the last few years. The system modeling and theoretical performance analysis of these systems have been well investigated and revealed in many studies, yet the system prototype and the corresponding experimental results are scarce. In this study, measurements of a 1 × 2 × 1 UHF passive RFID system, including a MIMO UHF passive RFID tag prototype and its corresponding software-defined radio-based reader, taken in a microwave anechoic chamber, are presented. The experimental results are compared with theoretical values and computer simulations. The overall results demonstrate the consistency and the feasibility of UHF MIMO passive RFID systems.

## 1. Introduction

Radio frequency identification (RFID) is a wireless communication technology that allows an object to be automatically identified through a unique identification code. The RFID systems are classified as passive when the tag has no battery, active when the tag has a battery, and semi-passive when the tag has a battery but still communicates with a reader using backscatter. A passive tag operates based on the principle of backscattering, in which an unmodulated signal is sent from a reader, and then reflected by the tag as a modulate signal. It takes advantage of RF energy harvesting, wireless power transfer, and the backscattering principle [1]. The ultra-high frequency (UHF) passive tags operate at 860–960 MHz, 2.45 GHz, and 5.8 GHz with the range distance in the order of 10 m [2].

The author in [3] presents an expandable wireless sensing platform designed to exhibit similar functionality to high performing RFID devices such as the Intel WiSP while remaining accessible as a learning tool.

Multiple input multiple output (MIMO) RFID systems are being widely investigated. The advantage of the multiple antennas RFID technologies over the single antenna RFID is that they allow many potential applications in the areas of accurate tracking and identification, since they are able to significantly increase the reliability and throughput [4]. On one hand, passive RFID tags present an advantage of being very small as compared with other communication devices, and therefore they are suitable to be used in very sensitive fields such as medicine. On the other hand, their small dimensions prevent these tags from incorporating some technologies that could enhance their operative performance, such as multiple antennas. The authors in [5] describe a developed analog 2 × 2 MIMO front end for an RFID rapid prototyping system which allows for various real-time experiments to investigate MIMO techniques such as beamforming, diversity combining, or localization at the reader side. In [6], an experimental evaluation of an UHF MIMO RFID system for positioning is presented, in which the authors propose a set of novel parametric maximum likelihood (ML) direct positioning algorithms capable of exploiting the coherent measurements performed by closely spaced antennas and simultaneously exploiting the non-coherent measurements by widely spaced antennas. In [5], the proposed MIMO concept enabled transmitter-sided digital beamforming after a measurement had been executed. An algorithm was presented to decompose the MIMO response to match a digital beamforming input matrix. A 2.4 GHz experimental reconfigurable localization system based on FMCW technology was set up to demonstrate the concept.

The theoretical performance analysis of RFID systems has been investigated in many studies The results in [4] can provide useful guidance on designing an RFID system with multiple antennas. The authors analytically study the BER performance of the MIMO RFID channel under two transmission schemes, i.e., the identical signaling transmission scheme and the orthogonal space-time coding (STC) scheme.

The moment generating function (MGF) approach has been used to obtain the average expression for the theoretical BER in M-PSK, -AM, or -QAM signals [2,4,7,8,9]. It is important for an analysis to test different channel conditions in order to understand the performance of any real system. Hence, a unified approach to evaluating the error-rate performance of a digital communication system operating over a generalized fading channel is presented in [10].

In recent studies, RFID technology has been incorporated into other modern technologies, such as in [11] where machine learning models were utilized to detect the direction of passive RFID tags. It has also been considered to be a major enabler of the Internet of Things (IoT) by [12], as new designs can easily integrate any type of sensors such as temperature, humidity, and water level sensor in smart homes technology [13].

For analysis and improved understanding of the UHF MIMO RFID system concept, we built a set of two devices, i.e., a tag equipped with two antennas and a reader with one transmit and one receiver antenna. Therefore, in this study, we analyze the bit error rate performance of a dual-antenna UHF passive RFID tag with polarization-time coded backscatter diversity proposed in [14] and later the corresponding simulations and measurements. Our main contributions in this study are to bring real measurement results of the so-called UHF MIMO RFID system not yet revealed in previous studies. The experimental results were compared with theoretical and simulated ones presented by several previous studies. We also demonstrate mathematically that our system model is a special case of that in [15].

This paper is organized as follows: in Section 2, we describe the system model and later the mathematical analysis; in Section 3, we describe the experimental verification of the proposed UHF MIMO RFID system; and in Section 4, we provide concludes for this study.

## 2. System Model

In RFID systems, both one-way and two-way channels are used to communicate, as shown in Figure 1. These are the so-called forward and backward channels. The 1×L×1 notation is used to represent an RFID system with one reader transmitting antenna, L tag antennas, and one reader receiving antenna. In [16], the system was alternatively called multiple input multiple output (MIMO).

Let us consider a system in which both channels are Gaussian. The expression to describe the whole channel can be represented as a composition of the forward and backscatter links, i.e., the overall channel is no longer Gaussian but the composition of the two channels. This channel is known as the dyadic backscatter channel (DBC), which was first introduced by [17].

The mathematical description of a 1×L×1 RFID system, considered as a special case of that in [15], is given as:(1)y=EhbDfG+n
where y∈ℂ1×J is the received signal vector, hb∈ℂ1×L is the vector containing the backward channel elements, G∈ℂL×J is the impedance modulated codeword drawn from a codebook 𝒞; sent by the tag antennas, n∈ℂ1×J is the vector containing the additive white Gaussian noise (AWGN) elements, and the matrix Df∈ℂL×L is built as follows:(2)Df=diag{hfxJ}=diag{hf1J}

In which hf∈ℂL×1 is the vector containing the forward channel elements. The vector xJ∈ℂJ contains the signal a priori known and sent by the reader transmitting antenna at the *J*-th time slot. The transmitting signal E represents energy and is considered to be constant at all-time instants. In this study, we implement and analyze the 1 × 2 × 1 RFID system. Recall that this system has one reader transmitting antenna, two tag antennas, and one reader receiving antenna.

### 2.1. Bit Error Rate (BER) Analysis

For the BER evaluation of radio frequency (RF) backscattering channels, we use the MGF as it transforms a more complex mathematical structure into a more tractable expression. Once the number of tag antennas is *L* = 2 and taking two time slots, the backscatter coding matrix is given by $G=\left(\begin{array}{cc} s_{0} & s_{1}^{*}\\ s_{1} & -s_{0}^{*} \end{array}\right)$. If **G** is transmitted, using binary phase shift keying (BPSK) modulation there are three possible codewords as follows:(3)$$\left(\begin{array}{cc} s_{0} & s_{0}^{*}\\ s_{1} & -s_{1}^{*} \end{array}\right)\;,\;\left(\begin{array}{cc} s_{1} & s_{0}^{*}\\ s_{1} & -s_{0}^{*} \end{array}\right)\;,\;\left(\begin{array}{cc} s_{0} & s_{1}^{*}\\ s_{0} & -s_{1}^{*} \end{array}\right)$$

The Hamming distance is given by the codeword difference from the received symbol (one of the above) and the true transmitted symbol. Normalizing the Hamming distance for the transmitted codeword $G^{u}$ and received $G^{v}$ we have $||g^{u}-g^{v}||\;=1$. For the quadrature phase shift keying (QPSK), the BER is defined as:(4)$$P_{e}=Q\left(\sqrt{g||\Upsilon |{|}_{F}^{2}E_{b}/N_{0}}\right)$$
where *Q*(*·*) is the *Q* function, *E**_b_* is the bit energy, and *g* is the constant gain that depends on the modulation scheme. For the QPSK modulation, *g* = 2 and we express *γ* = *E_b_/N*_0_. The BER of our backscatter system, averaged over the channel $\Upsilon =h^{b}D^{f}$ is given by:(5)$$\begin{array}{l} p_{e}\left(G^{u}\to G^{v}|\Upsilon \right)\leq \frac{1}{2}\exp||\Upsilon |{|}_{F}^{2}\gamma \\ p_{e}\left(G^{u}\to G^{v}|\Upsilon \right)\leq \frac{1}{2}\exp\left(-\sum_{i=1}^{L}\gamma \delta_{i}\rho_{ii}\right) \end{array}$$

Because the two random variables are independent, we can write:(6)$$\begin{array}{l} E_{\delta ,\rho }\left[\frac{1}{2}\exp\left(-\sum_{i=1}^{L}\gamma \delta_{i}\rho_{ii}\right)\right]\\ =\frac{1}{2}\prod_{i=1}^{L}E_{\delta_{i}\rho_{ii}}\left[\exp\left(-\sum_{i=1}^{L}\gamma \delta_{i}\rho_{ii}\right)\right] \end{array}$$

In a real propagation environment, the line-of-sight (LoS) component is present and more predominant. Therefore, the Rician channel model has been chosen to represent forward and backward channels.

We consider two antennas attached to a tag with vertical and horizontal polarizations, therefore, the channel vectors $h^{b}$ and $D^{f}$ become uncorrelated Rician random variables as in [8].

Then, the final expression is given by:(7)$$p_{e}(\bar{\bar{\gamma }},L)=C_{L}{\left(C_{1}^{f}C_{2}^{b}\right)}^{L}{\left(\left(\ln\left(\bar{\gamma }\right)+C_{3}^{fb}\right){\bar{\gamma }}^{-1}\right)}^{L}$$
where $\bar{\bar{\gamma }}$ is the MGF of $||\Upsilon |{|}_{F}^{2}$, $C_{L}=(\Gamma ((1/2)+Lb)/2\sqrt{\pi }\Gamma (1+Lb))$, $C_{1}^{f}=K_{2}e^{-K_{f}}$, $C_{2}^{b}=K_{1}e^{-K_{b}}$, $C_{3}^{fb}=e^{K_{f}}-1-\ln(K_{1}K_{2})$, $K_{1}=K_{b}+1$, and $K_{2}=K_{f}+1$. The parameters $K_{f}$ and $K_{b}$ characterize the Rician channel and are for forward and backward channels, respectively. Optionally, and for a more tractable system behavior analysis, the Nakagami-m expression is an alternative. Similarly, for the Nakagami-m, the parameters *m_f_* and *m_b_* characterize the forward and backward Nakagami-m channels, respectively.

The diversity gain of our system is given as in [9] and is *L = 2*:(8)d=LN−L(N−1)=L=2
where *d* is the diversity gain and *N =* 1 is the number of the reader receiving antennas.

### 2.2. UHF MIMO RFID System Architecture

#### 2.2.1. UHF MIMO RFID Tag Architecture

##### Front-End and Tuning

The schematic of a UHF MIMO RFID tag analog circuit is shown in Figure 2 and was built with reference to WISP tag design [7]. The circuit structure is composed of two front-end antennas, matching circuit, charge pump circuit, demodulation circuit, voltage regulator/rectifier, and back-end signal processing module. The UHF MIMO RFID tag has relatively high power consumption as compared with the single antenna one, therefore, the rectifier is designed to deliver more current than ordinary tags. The circuit is excited by a commercial EPC Class 1 Generation 2 compliant reader, operating at 902–928 MHz with allowable transmission power of 4 W_EIRP_ (effective isotropic radiated power) [7].

The front-end part of the tag is composed of two antennas shifted 90° (with polarization directions orthogonal to each other) as for vertical and horizontal antenna polarization. This design was made in order to achieve independent channel features. Each antenna has two load impedances, $Z_{i,k}$ representing the impedance k responsible for the *i*-th antenna as in Figure 2. The five-volt doubling circuit is used for collecting enough energy to activate the integrated circuit (IC). Low threshold RF Schottky diodes are used to maximize the output voltage of the rectifier. Finally, a large capacitor stores the rectified direct current (DC) voltage and supplied it to a 1.8 V regulator. Then, this voltage is used to power the MSP430 (manufactured by TEXAS INSTRUMENTS family). The matching circuit of the passive UHF MIMO RFID tag, composed of a series inductor and parallel trimmable capacitor, is tuned until the output voltage of the tag is maximized. The power harvester is a nonlinear device with a load dependent efficiency.

The front end must be tuned to provide maximum output voltage in the presence of the desired load.

In the design, Stage 1 of the charge pump is replaced by the transistor HSMS-285C to achieve high performance and an optimized size. The matching part is fine tuned in the last step after collecting the latest tag impedance. Finally, the impedance matching is implemented to obtain 50 Ohm with the optimized transmission line. When operating in the forward link (reader-to-tag), the matching impedances maximize the power received by the tag and in the backward link (tag-to-reader), the tag impedance (controlled by the tag) generates the desired backscatter signal.

##### Modulation and Demodulation

To encode reader-to-tag data for transmission, the reader amplitude modulates the 925 MHz RF carrier wave. The carrier waveform remains at a constant amplitude until the bits are transmitted, making the carrier drop nearly 10% of its normal value. Logical “one” and “zero” are indicated by the “break” duration. The harvester leaves the baseband data signal in the order of 70 kHz and effectively demodulates the 925 MHz carrier signal. Most of this signal is rectified into DC voltage. When the demodulator is enabled, some of the signal travels through the demodulator branch. This signal is also rectified to produce a reference voltage for the bit detection process.

The ten (10) diodes in the voltage doubling ladder are the RF Schottky diodes. Instead of using a second mini harvester for demodulation, the demodulator is directly connected to the main harvester; however, it can be disabled to prevent energy leaking from the demodulation’s pull-down resistor. In Figure 3, LS is the lever shifter.

The final extra diode is responsible for an additional rectification step, in which the 70 kHz data signal is removed and leaves a slowly varying average power level that provides a dynamic reference for bit detection. Through a Schmitt trigger inverter and level shifter, the 70 kHz data signal is fed to convert the relative magnitude of the incoming data waveform into a 1.8 V level for the MSP430. The ASK demodulation is used. The NLSVT244 is a 2-bit configurable dual-supply voltage level translator by ON Semiconductor^®^. The typical applications are in mobile phones, PDAs, and other portable devices. It has the same function with bit detection.

The ultra-low power microcontroller MSP430 has all the general purpose computation abilities of the UHF MIMO passive RFID tags. It is responsible for many core functions such as demodulating reader signals and controlling backscatter signal modulator. These passive tags modulate the impedance of their antenna, which causes a change in the amount of energy reflected back to the reader, rather than actively transmitting radio signals. This modulation is the so-called backscatter modulation. The ADG918 switch, manufactured by ANALOG DEVICES, that connects two different loads is used to change the impedance matching of the corresponding antenna. When the MSP430 is activated via the harvester, the IC will run the algorithm to control the RF switch ADG918. Simultaneously, the IC sends the control signal to the protection components (including the R, C, and diode) to short circuit the two branches of the antenna, changing the antenna impedance state.

##### Digital Section and Power Consumption

Considering the design of passive tags, power consumption minimization is always required. This minimization is reached through careful component selection as the power available to feed the RFID is extremely limited. Benefiting from the fact that IC manufacturing now allows for the design of discreet components with less than 1 uA current consumption and 1.8 V operation, it is now possible to construct working passively powered RFID tags with discrete components.

The 16-bit flash microcontroller used in the MSP430F1232 can run at up to 4 MHz with a 1.8 V supply voltage and consumes approximately 470 uA when active for this choice of frequency and voltage. In the design, the tag readable range has been extended by improving the microcontroller’s firmware, allowing the operation at lower voltage and clock frequency. Of particular interest for low power RFID applications is the MSP430 that has various low power modes, and the minimum RAM-retention supply current of only 0.5 uA at 1.5 V. The device provides over 8 Kbytes of flash memory, 256 bytes of RAM, and a 10-bit, 200 kilo samples per second analog-to-digital converter (ADC).

The processor uses selective combining techniques to select the signal-to-noise ratio (SNR).

Table 1 synthesizes the design comparison between the proposed UHF MIMO RFID tag, the WISP design [7], and the UHF MIMO RFID tag in [14].

#### 2.2.2. UHF MIMO RFID Reader Architecture

The National Instruments (NI) NI-5644R was used to build the UHF MIMO RFID reader. Two patch antennas are connected to the input and output RF ports. One antenna performs as the transmitter and the other as the receiver. We used the NI-5644R vector signal transceiver (VST) which includes the VSG and VSA, the vector signal analyzer and the vector signal acquisition, respectively.

##### System Protocol

For establishing communication, first the reader sends a query command and a continuous carrier wave. Considering the tag energy consumption problem, the query is set as a negative trigger signal. After the tag successfully recognizes the negative trigger, the backscatter signal is changed by changing the frontend impedance. The data format of the backscatter signal is as shown in Figure 4.

The tag’s backscatter signal is built as a response from the two tag antennas. As can be seen in Figure 4, there are the pilot and tag UID signals. The purpose of this backscatter signal is to let the reader synchronize the signal and estimate the channel state information (CSI). A more detailed explanation can be found in our previous study. The received signal strength indicator (RSSI) is cancelled out and the unique identification (UID) code is given by the six levels. In this case, the UID code was set to [011010].

The baseband signal processing architecture is shown in Figure 5. It is mainly divided into four blocks: the DC carrier estimation, synchronizer, channel estimator, and decoder.

##### DC Carrier Estimator

The DC Carrier Estimator is responsible for estimating the DC component of the carrier received by the reader receiving antenna and eliminating it to keep the tag signal of interest. Even when the tag backscatters the signal, the reader will continue transmitting the carrier in order to provide energy to the tag. Therefore, the RF signal received by the reader includes the carrier energy leaked from the transmitting antenna in addition to the tag backscatter signal. We use the method of time domain for estimation of carrier information to eliminate the carrier signal. The discrete I (in phase) and Q (quadrature) information are only extracted at the time of the DC component. Later, its average value is calculated, that is, the DC carrier signal is estimated, and the tag backscatter signal is deducted from the estimated DC signal.

##### Channel Estimator

The corresponding channel estimation result and the maximum ratio combining technique are used to obtain the demodulation signal. The purpose of channel estimation is to obtain channel information that could cause signal attenuation in a wireless communication environment, mostly unknown phase changes and amplitude variations. We assume that it is a non-time-varying channel environment and we use the least square method (LS) for estimation [18]. Since our tag is manually soldered it is susceptible to phase and gain errors. This effect is shown in Figure 6, where $\Gamma_{L,i}$ indicates the *i*-th ideal reflection coefficient of the *L*-th tag antenna, and ${\tilde{\Gamma }}_{L,i}$ indicates the *i*-th measured impedance reflection coefficient of the *L*-th tag antenna. To reduce errors, measurements are taken beforehand, and then compensated during the estimation.

To measure the reflection coefficient offset, the tag was connected to the network analyzer, then, we obtain the value of ${\tilde{\Gamma }}_{L,i}$. The values of the measurements previously taken in our lab are shown in Table 2.

##### Decoder

Now, the maximum likelihood detection technique and Miller decoding are performed. After the CSI by the channel estimation, the reader receiving antenna uses its respective synchronization starting positions to obtain the starting positions of the received UID discrete signals, ready to be decoded by STBC. When the correlation is completed, the decoding result is the UID code (011010), as shown in the following Figure 7:

## 3. Experimental Verification of the Proposed UHF MIMO RFID System Performance

A dual antenna UHF MIMO RFID tag operating at 925 MHz built in our laboratory was used for the experiments. Figure 2a shows the UHF RFID MIMO tag circuit architecture and a photograph of it is shown in Figure 2b which is designed with reference to WISP tag [7]. The part of the front-end antenna is composed of Antenna 1 and Antenna 2 with their polarizations orthogonal to each other. As previously mentioned in this study, this design allows the tag to exhibit uncorrelated channel characteristics when transmitting or receiving signals, achieving the effect of polarization diversity. In the forward link, there are two sets of symmetric modules transmitting signals to the backend processor MSP430. This processor uses selective combining techniques to select the SNR. In order to achieve the passive effect of the tag, the ADG918 component [19] was used in the front-end RF switcher.

The reader is as specified in the previous section. In NI-5644R from National Instruments the LabView is installed and was used to control the field-programmable gate array (FPGA) that controls the vector signal transceiver (VST).

The experiments were taken at the Anechoic Chamber of National University of Science and Technology (NTUST), whereas good approximation to the Gaussian channels could be reached. The reader transmitting and receiving antennas, Tx and Rx, respectively in Figure 8, are both unidirectional patch antennas with a gain of 6 dBi. The antennas use circular polarization.

The tag antennas are omnidirectional (approximated). More details about the tag can be found in [14]. It is worth mentioning that the tag built in our laboratory and used for the experiments is unique, in the sense that it performs backscatter modulation with a similar principle to that proposed in [15], but with its particular advantages earlier and well introduced by our laboratory group in [14].

As can be seen in the Figure 8 and Figure 9, the separation between the tag and the reader antennas was set to *d* = 1.8 m (meters). The interference can be ignored inside the chamber. Assuming that the noise in the non-reflective environment is very small, the only considered and measured noise is the thermal noise generated by the instrument. As a strong LoS component is present, Rician better represents the chamber room environment. As used in theory, *K_f_* and *K_b_* represent the Rician channel parameters, which are for forward and backward channels, respectively [20].

To obtain the simulation results, we set the reader with a carrier frequency of *f_c_ =* 925 MHz, the number of backscatter bits as 1,000,000 in total at each SNR level. The number of Monte Carlo simulations is *N =* 50,000. To perform this experiment, in the first place, the reader is allowed to repeatedly transmit query instructions 50,000 times, so that the tag continuously returns the backscattered signal and decodes it. The decoding part is divided into maximum likelihood detection decoding and Miller decoding. The error accumulation is the sum of the symbol errors, latter the sum of bit errors, which are then used to calculate the SER and BER, respectively. To obtain different values of SNR, we added FAT-AM5AF5G6G2W3 attenuators at the receiver side (connecting to the VST), with a gap of 3 dB in each experiment trial. We used a 50 Ω terminator to obtain the thermal noise for the SNR reference value. For the channel variance, we used 0 dB. Each antenna of the tag uses the BPSK modulation as in the mathematical description. The key parameter *K_f_* and *K_b_* are illustrated later.

When the values of *K_f_* and *K_b_* are less than one, the worse channel case is obtained. When *K_f_ = K_b_* = 1, the equivalent to a Rayleigh channel can be obtained. In Figure 10, the Rayleigh equivalent curve is plotted and as compared with our results we can see a considerable gap, which proves that the Rayleigh channel does not characterize either our system or an indoor environment. When *K_f_* and *K_b_* tend to infinity, a channel that is more approximated to the AWGN channel curve can be obtained [20].

It is worth mentioning that, in our case, the values of *K_f_* and *K_b_* should be greater than one. In Figure 10, Figure 11 and Figure 12, the AWGN equivalent curve is plotted. The AWGN case is the ideal one, hence, we have used the Rician channel model to obtain results that fit with our real experiments. The theoretical results curve tends to highly change behavior with adjustments of the Rician channels parameters, which is clearly seen in Figure 11 and Figure 12. In Figure 11 the values of *K_f_* and *K_b_* are both equal to four.

More results can be obtained by also adjusting the values of *K_f_* and *K,* as in the Figure 12, where *K_f_ =* 4.5 and *K_b_* = 4. Here, we sat *K_f_* different to *K_b_*, i.e., the forward and backward links are characterized by different Rician channel parameters. Both simulation and experimental results are consistent, as their curves tend to have the same behavior which can be seen in Figure 11 and is obvious in Figure 12.

## 4. Conclusions

In this study, we verified the theory and simulations studied in previous studies through measurements for 1×L×1 MIMO systems. We analyzed a real 1×2×1 passive UHF MIMO RFID system with a tag built in our laboratory. The experimental results show consistency with the theoretical results as well as the simulations. The links were both considered to be Rician with the *K* parameters which tend to a Gaussian channel condition as the used environment is the anechoic chamber.

## Figures and Tables

**Figure 1 sensors-21-06308-f001:**
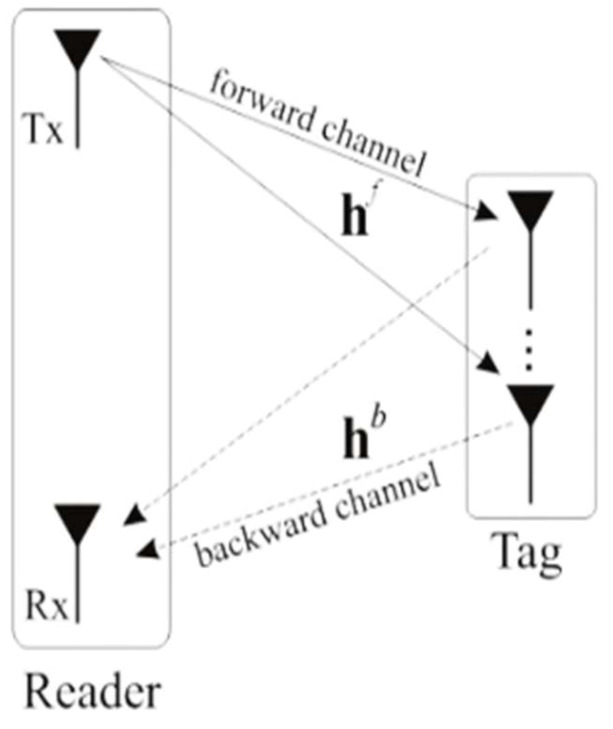
The backscatter system configuration.

**Figure 2 sensors-21-06308-f002:**
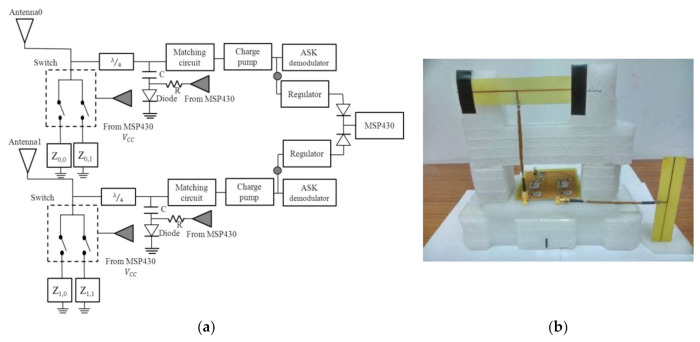
MIMO UHF RFID tag: (**a**) Architecture; (**b**) photograph.

**Figure 3 sensors-21-06308-f003:**
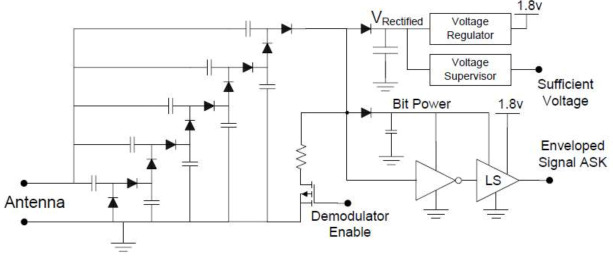
Schematic of harvesting (**left**) and ASK demodulation (**right**) circuit.

**Figure 4 sensors-21-06308-f004:**
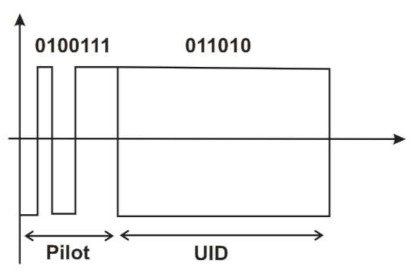
MIMO tag backscatter signal format.

**Figure 5 sensors-21-06308-f005:**
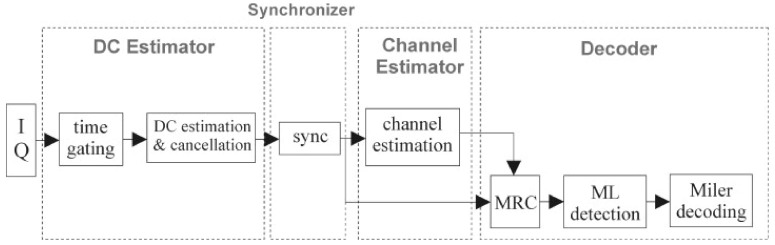
Baseband signal processing architecture (reader).

**Figure 6 sensors-21-06308-f006:**
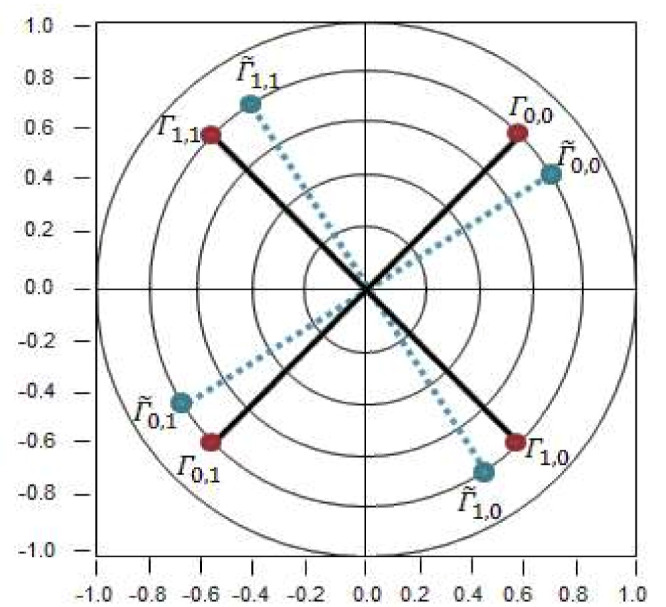
Backscatter constellation.

**Figure 7 sensors-21-06308-f007:**
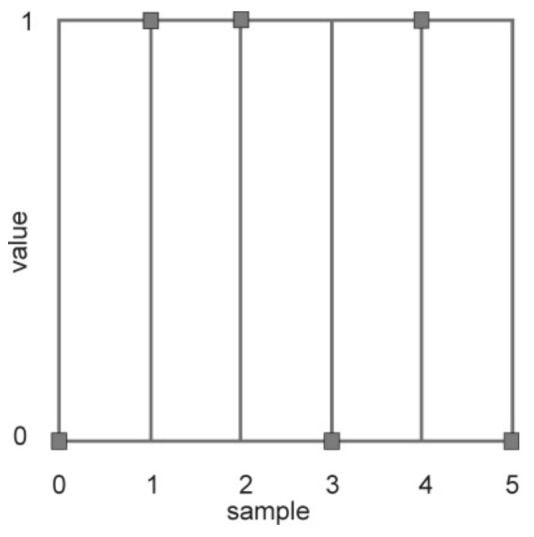
Miler decoder, UID code.

**Figure 8 sensors-21-06308-f008:**
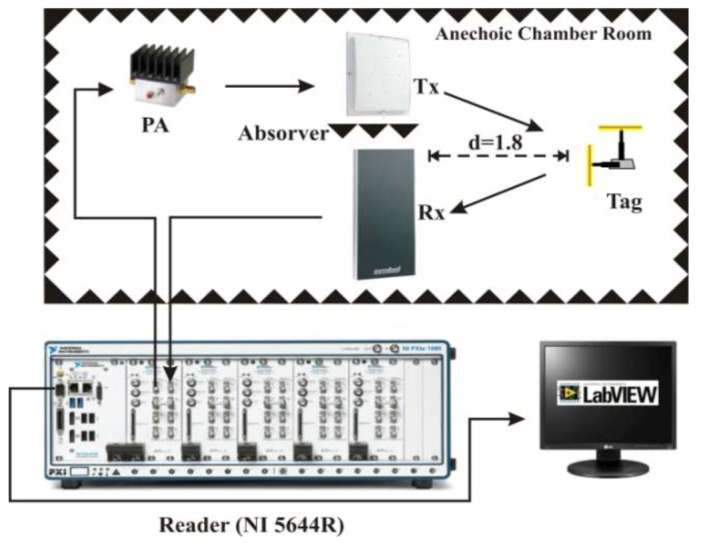
Experimental setup.

**Figure 9 sensors-21-06308-f009:**
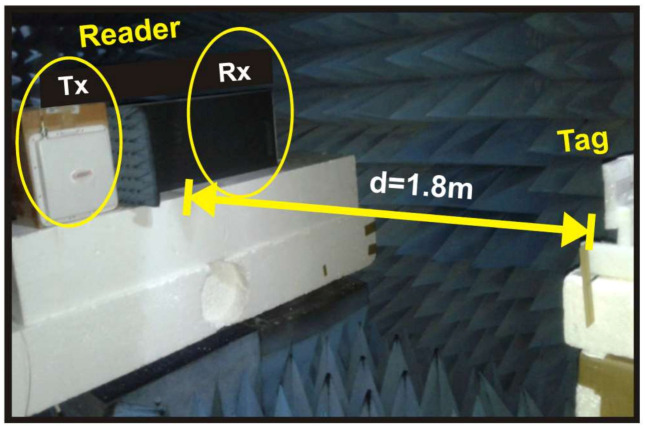
Photograph of the 1 × 2 × 1 system displacement in the NTUST anechoic chamber.

**Figure 10 sensors-21-06308-f010:**
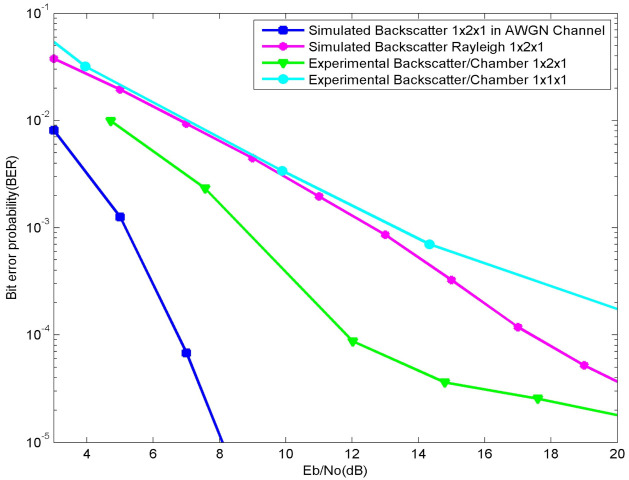
Plot of the experimental and simulated BER results of a 1 × 2 × 1 system vs. a 1 × 1 × 1 system as compared with the AWGN and Rayleigh.

**Figure 11 sensors-21-06308-f011:**
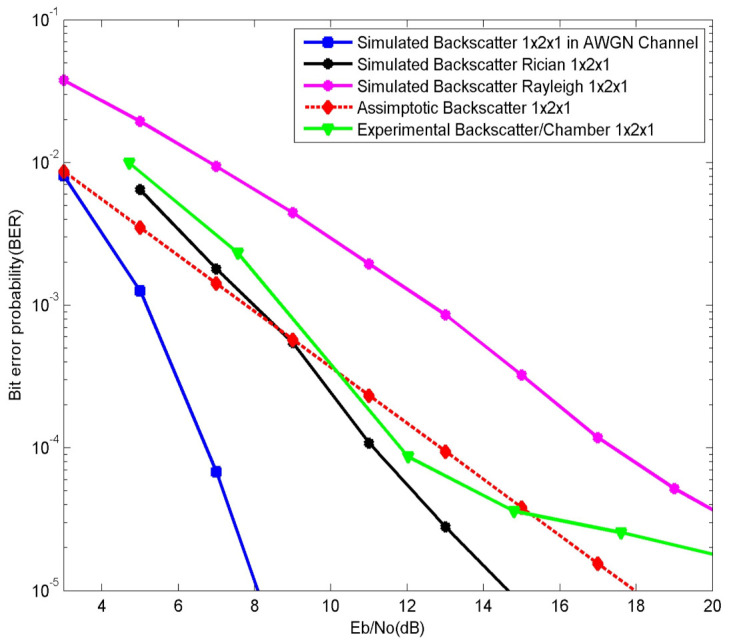
The BER results of a 1 × 2 × 1 system. Theoretical, simulated, and experimental results, *K_f_* and *K_b_* = 4.

**Figure 12 sensors-21-06308-f012:**
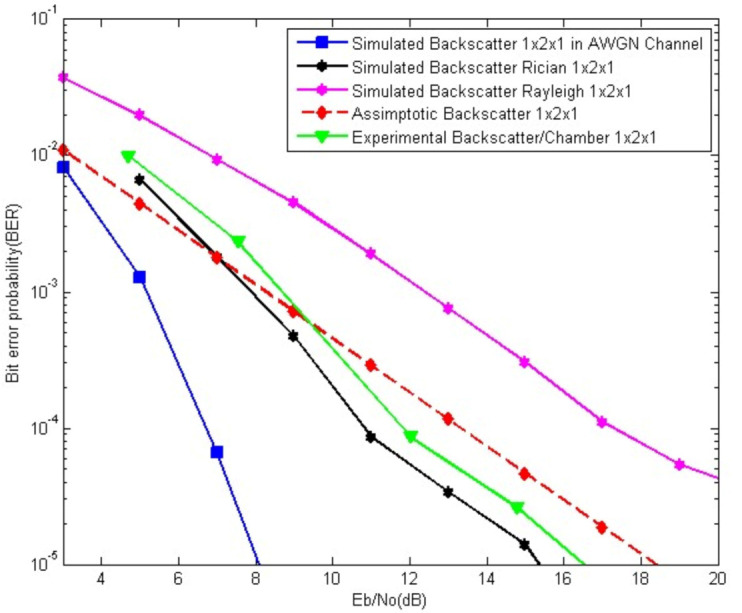
The BER results of a 1 × 2 × 1 system. Theoretical, simulated, and experimental results, *K_f_* = 4.5 and *K_b_* = 4.

**Table 1 sensors-21-06308-t001:** WISP and UHF MIMO RFID TAG RF design comparison.

	UHF MIMO RFID TAG	WISP TAG	MIMO RFID TAG in [14]
Receiver			
Transistor	HSMS-285C	HSMS-285C	No
Regulator	NCP583	NCP583	Yes
Bit translator	NLSV2T244	NLSV1T244	Yes
	S100C30N4T		
Matching	lamda/4 and R&C tunning	R&C and 50 Ohm transmission line	R&C tuning
Super cap	Yes	Yes	No
Tempertature, accelator sensor,3D axis sensing	No	YesADXL330, TS5A3166, LM94021	
EEPROM	No	24AA08	Yes
Microcontroller			
MCU	MSP430F1232	MSP430 × 21 × 2	MSP430F1232
Crystal	No	FC-135	No

**Table 2 sensors-21-06308-t002:** Tag reflection coefficients at 925 MHz.

Reflection Coefficient	Tag Values
${\tilde{\Gamma }}_{0,0}$	0.4∠−131.9°
${\tilde{\Gamma }}_{0,1}$	0.7∠37.69°
${\tilde{\Gamma }}_{1,0}$	0.35∠−17.81°
${\tilde{\Gamma }}_{1,1}$	0.71∠130.34°

## Data Availability

Not applicable.

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
