# Peer review of "Experimental Results and Performance Analysis of a 1 × 2 × 1 UHF MIMO Passive RFID System"

_sensors, 2021, doi:10.3390/s21186308_

Round 1

Reviewer 1 Report

The demonstration of the 1x2x1 UHF passive RFID system scheme including a MIMO UHF passive RFID tag prototype with the corresponding software-defined radio-based reader shows the feasibility of the system well.  However, several points requiring further explanations as listed below.

- The necessity and novelty of this work have not been clearly presented in the introduction. In the current form, it is said that 1x2x1 structure is a robust structure, so BER and other analyses are presented in the work.  It may seem weak.

- Many practical RFID wireless channels, the system noise has non-AWGN statistics. How would it affect the system model and simulation results?

- In Fig. 10-12, the experimental backscatter/chamber 1x2x1 case shows an error floor. There’s an insufficient explanation for it. Adding a more straightforward explanation is needed.

- There has to be a proper reference for “ For insightful analysis and better understanding on the concept of the UHF MIMO RFID system, a set of a tag equipped with two antennas and a reader with one transmitting and one receiving antennas has shown to be a powerful tool.”

- Some corrections in the text are required for a better understanding. Your sentence may be unclear or hard to follow.  Just to name of a few, line 35: “notorious advantage” and line 39: “devices what makes them suitable to be used.”

Reviewer 2 Report

Introduction: The authors should give a more detailed description of the novelty and the advantages of the proposed work to existing literature.

Page 5, line 175: “In the design, stage 1 of the charge pump is replaced by the transistor HSMS-285C …” a figure showing the design of the charge pump would be helpful

Page 5, line 167, 175, 210, 361: MSP430, HSMS-285C, ADG918, FAT-AM5AF5G6G2W3 -> the manufacturer is missing

Page 6, line 194: type of Schottky diodes and manufacturer

Page 6, line 203: What is NLSVT244 ?

Page 6, line 210: It is probably meant that the switch for the modulation connects 2 different loads to the antenna

Page 6: It is stated that the power consumption of the uC at 4MHz is about 0.5mA which is quite a lot for RF powering. Was the maximum powering distance of the RFID tested? A diagram of the available current from the harvesting circuit over the distance would be helpful.

Chapter 3: Which modulation format was used for the experiments and the simulations?

Round 2

Reviewer 2 Report

no further comments